# Coherent control of enhanced second-harmonic generation in a plasmonic nanocircuit using a transition metal dichalcogenide monolayer

Pei-Yuan Wu[1], Wei-Qing Lee[1], Chang-Hua Liu [1] ✉ & Chen-Bin Huang [1] ✉

Nonlinear nanophotonic circuits, renowned for their compact form and integration capabilities, hold potential for advancing high-capacity optical signal processing. However, limited practicality arises from low nonlinear conversion efficiency. Transition metal dichalcogenides (TMDs) could present a promising avenue to address this challenge, given their superior optical nonlinear characteristics and compatibility with diverse device platforms. Nevertheless, this potential remains largely unexplored, with current endeavors predominantly focusing on the demonstration of TMDs' coherent nonlinear signals via free-space excitation and collection. In this work, we perform direct integration of TMDs onto a plasmonic nanocircuitry. By controlling the polarization angle of the input laser, we show selective routing of second-harmonic generation (SHG) signals from a $MoSe_2$ monolayer within the plasmonic circuit. Routing extinction ratios of 14.86 dB are achieved, demonstrating good coherence preservation in this hybrid nanocircuit. Additionally, our characterization indicates that the integration of TMDs leads to a 13.8-fold SHG enhancement, compared with the pristine nonlinear plasmonic nanocircuitry. These distinct features—efficient SHG generation, coupling, and controllable routing—suggest that our hybrid TMD-plasmonic nanocircuitry could find immediate applications including on-chip optical frequency conversion, selective routing, switching, logic operations, as well as quantum operations.

Plasmonic nanocircuits simultaneously offer ultrahigh operating frequency up to the visible range while maintaining ultrasmall footprint down to the nanometer regime. Despite the impressive results demonstrated by linear plasmonic circuits[1], recent research trend is devoted to further expand the circuit functionalities through the extension into nonlinear regimes[2–4]. Nonlinear photonic circuits could offer even higher operational frequencies and enhanced light-matter interactions, providing unique advantages for on-chip optical frequency conversions and quantum signal processing[5–10].

Nonlinear plasmonics are motivated by two main attributes of surface plasmons: (1) large local field enhancement; and (2) automatic symmetry breaking at the metal-dielectric interface. These traits suggest plasmonic structures would be ideal candidates to achieve even-order frequency up-conversions despite typical noble metals being centrosymmetric. Various plasmonic devices have been designed to achieve second-harmonic generations (SHGs)[11]. However, the SHG efficiencies obtained in past plasmonic systems remain low for practical applications. To address this challenge, various approaches have

[1]Institute of Photonics Technologies, National Tsing Hua University, Hsinchu, Taiwan. ✉e-mail: chliu@ee.nthu.edu.tw; robin@ee.nthu.edu.tw

been proposed to enhance the SHG efficiency in plasmonic circuits, including waveguide geometry optimization and the incorporation of nonlinear optical materials to form hybrid plasmonic platforms[3,12–15].

In recent years, transition metal dichalcogenides (TMDs) have attracted significant attention due to their unique electronic and optical properties[16]. One of the intriguing properties of TMDs is their strong SHG capability, which arises naturally from the lack of inversion symmetry in their crystal structure. Compared to conventional optical materials, TMDs exhibit one order of magnitude higher effective $\chi^2$ value, making them ideal for nonlinear optical applications[17]. Moreover, due to the strong in-plane covalent bonding and weak van der Waals interlayer interaction[18], TMDs can be easily integrated onto other device platforms[2,19–21]. This unique characteristic makes TMDs promising materials for integrated photonics, a capability not offered by conventional bulk materials. However, it is worth noting that TMD-generated SHGs have, so far, predominantly been reported in a mechanism involving free-space excitation and free-space collection[12,22,23].

In this work, we experimentally demonstrate an efficient nonlinear hybrid photonic nanocircuit by integrating a TMD material onto a functional plasmonic nanocircuit. The SHG signals from a MoSe$_2$ monolayer are effectively coupled into the nanocircuit based on plasmonic two-wire transmission-line (TWTL)[14,24–27] architecture. Furthermore, the TMD-generated SHG polarization dependence is seamlessly mingled with the unique polarization-selective routing functionality of the plasmonic circuitry to enable coherent control. We experimentally achieve routing of the SHG signals when the input fundamental frequency laser is linearly polarized at ±11.25°. The

experimental routing extinction ratio can reach as high as 16.50 dB. Such polarization-selective routing is compared to both theoretical and numerical analysis and are found to be in perfect agreement. Furthermore, detailed quantitative analyses are performed to reveal our MoSe$_2$-plasmonic integration approach enhanced the SHG efficiency up to 13.8-fold. These results provide a promising avenue for achieving efficient coherent control of the SHG by TMDs, realizing their full potential in nonlinear integrated photonics.

## Results and discussion
### Circuit working principle
Figure 1 schematically illustrates the working principle of our hybrid nonlinear router: Fig. 1A shows that a linearly polarized SHG signal ($\mathbf{E}_{2\omega}$) is generated when the MoSe$_2$ monolayer is excited by a linearly polarized input fundamental frequency ($\mathbf{E}_\omega$) laser. When the input laser is polarized with an angle $\varphi_\omega$ with respect to the $x$-direction (defined parallel to one of the MoSe$_2$ armchair edges), the output SHG from MoSe$_2$ monolayer is polarized with an angle of $\varphi_\omega + 3\varphi_\omega$[28,29]. With an additional output polarization analyzer oriented in ±45° (the analyzer angles are related to the plasmonic router, to be addressed later) with respect to the $x$-direction, the transmitted SHG intensities ($I_{2\omega,\ 45°}$ and $I_{2\omega,-45°}$) could be described by the following equations (detailed in Supplementary Figs. 1 and 2):

$$I_{2\omega,45°} = \mathbf{E}_{2\omega}^2 \cos^2(\varphi_\omega + 3\varphi_\omega + 45) \tag{1}$$

$$I_{2\omega,-45°} = \mathbf{E}_{2\omega}^2 \cos^2(\varphi_\omega + 3\varphi_\omega - 45) \tag{2}$$

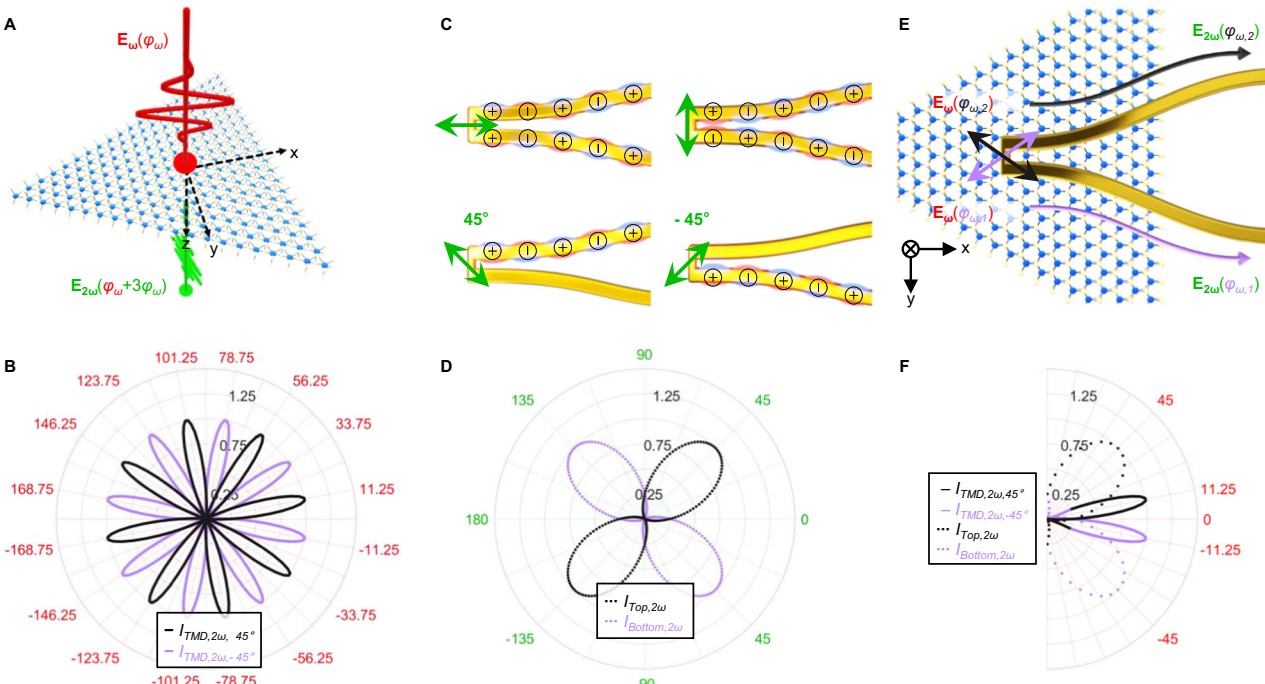

Fig. 1 | Working principle. A Schematics of second-harmonic generation (SHG) from the MoSe$_2$ monolayer. The MoSe$_2$ lattice is artistically presented: blue/yellow spheres represent molybdenum/selenium atoms, respectively. The MoSe$_2$ monolayer lies in the $x$–$y$ plane, excited by a fundamental frequency ($\mathbf{E}_\omega$) laser (red field arrow) propagating towards the $z$-direction. The SHG signal ($\mathbf{E}_{2\omega}$) from MoSe$_2$ is expressed by the green field arrow, where $\varphi_\omega$ is the angle between the polarization angle of the input laser and the $x$-direction (defined parallel to one of the MoSe$_2$ armchair edges). B Polar plots for the MoSe$_2$-induced SHG when a polarization analyzer is orientated in ±45°. C Schematic illustration of the polarization-selective modal properties of a plasmonic two-wire transmission-line (TWTL) router. When the input signal is parallel/perpendicular to the nanowire direction, the symmetric/

antisymmetric mode is excited. When the input laser field is 45°/−45° linearly polarized, only the top/bottom nanowire carries surface plasmon polariton (SPP) field, respectively, where the charge densities of each excited mode are represented by black circles with + and − symbols. D Polar plots of the router for second-harmonic signals. E Schematics of the functionality of the MoSe$_2$-plasmonic hybridized integrated nonlinear router. MoSe$_2$ is placed on top of the plasmonic router. Different routing directions of SPP fields are depicted with purple and black arrows corresponding to different linear polarization states of the input signal. F Polar plot for the routing period comparison of MoSe$_2$ (solid lines) and plasmonic (dotted lines) indicates the MoSe$_2$-plasmonic hybrid circuit routing could be achieved when $\varphi_\omega = \pm 11.25°$.

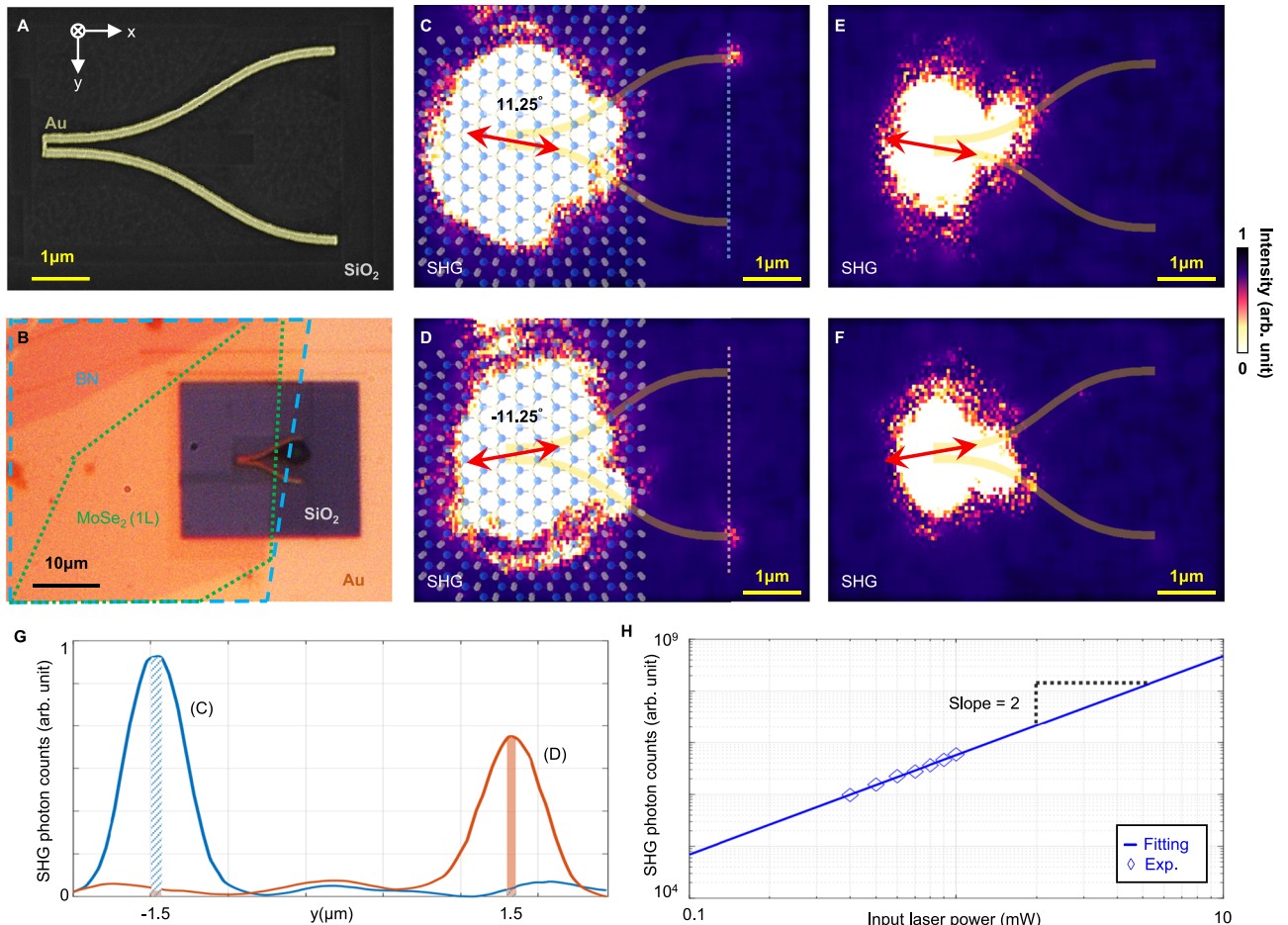

**Fig. 2 | Fabrication and performance of the MoSe₂-plasmonic hybrid router.**
**A** The SEM image of the fabricated plasmonic router. The dark region is SiO₂ and the bright region is gold. **B** The optical microscope image of the fabricated MoSe₂-plasmonic hybrid router. MoSe₂ monolayers with hexagonal boron nitride (hBN) protection are carefully aligned to the router input port. The blue dashed lines indicate the region covered by hBN, while the region outlined by green dashed lines presents MoSe₂. The yellow region is the gold film and the dark region is the SiO₂ substrate. **C** $2\omega$ SPPs are routed to the upper output port when the laser field is polarized in 11.25°. **D** $2\omega$ SPPs are routed to the bottom output port when the laser field is polarized in −11.25°. The MoSe₂ lattice is artistically presented in (**C, D**). **E, F**

No SHG signals can be observed for the reference pure plasmonic router. Since the distance between the two output ports is 3 μm, the scale of (**C–F**) can be determined by the top and bottom scattering points. **G** The SHG line intensity along the dashed line in (**C**) is plotted as the blue trace, and the orange trace is the result of (**D**), where the widths of blue and orange rectangles correspond to the region (130 nm width) of two output ports individually. **H** The log–log plot of the SHG photon counts for the MoSe₂-plasmonic hybrid router as a function of input laser power. The experimental data are presented as symbols. The blue line represents fitting results of the experimental data with polyfit function in MATLAB.

Figure 1B displays the variations of $I_{2\omega, 45°}$ and $I_{2\omega, -45°}$ as a function of $\varphi_\omega$, revealing that for each analyzer orientation, there would be eight input laser angles that give rise to maximum transmission[28,29].

Figure 1C depicts the working principle of our polarization-selective plasmonic TWTL router. A plasmonic TWTL comprises two identical metallic nanowires separated by a gap distance. The TWTL supports two surface plasmon polariton (SPP) modes that can be independently excited, solely determined by the input signal polarizations[24,25]. When the input signal is linearly polarized parallel to the wire direction, the symmetric mode is excited. On the other hand, the antisymmetric mode is excited when the input signal is polarized perpendicular to the wire direction. The symmetric mode has in-phase charge densities on the two nanowires, while the charge densities on the two nanowires are π out of phase for the antisymmetric mode. Our TWTL input antenna is designed to provide (1) a 1:1 amplitude ratio between the symmetric and the antisymmetric mode; and (2) a phase difference of 0 between the two modal fields. Such antenna design allows ideal superposition of the two modes for complete cancellation of the SPP fields on one of the two nanowires, which serves as the basis for polarization-selectively routing[25]. When the input signal is 45°

linearly polarized, only the top nanowire carries the SPP field. On the other hand, when the input signal is −45° linearly polarized, only the bottom nanowire carries the SPP field. Figure 1D shows the polar form of selective routing angular periodicity in the plasmonic router is $\varphi_\omega = \pm45°$. Details of the TWTL design are provided in the Supplementary Fig. 4.

By integrating the MoSe₂ monolayer onto the plasmonic TWTL, Fig. 1E illustrates the functionality of the anticipated hybrid polarization-selective nonlinear router. The fundamental laser polarization angle $\varphi_\omega$ is used to serve as the deciding turn-key for the routing direction of the MoSe₂ SHG signal upon conversion into SPP. For this hybridized integrated circuit, it is expected that perfect routing of the SHG signal will be achieved at $\varphi_\omega = \pm11.25°$ (±45° divided by four), as indicated in Fig. 1F. Details to the mathematical expression are provided in Supplementary Figs. 1 and 2.

**Experimental selective routing confirmation**
Our plasmonic nanocircuit is fabricated using focused-ion beam milling (FEI Helios) on an electron-beam-evaporated gold thin film with 60 nm thickness. The scanning electron microscope (SEM) image of

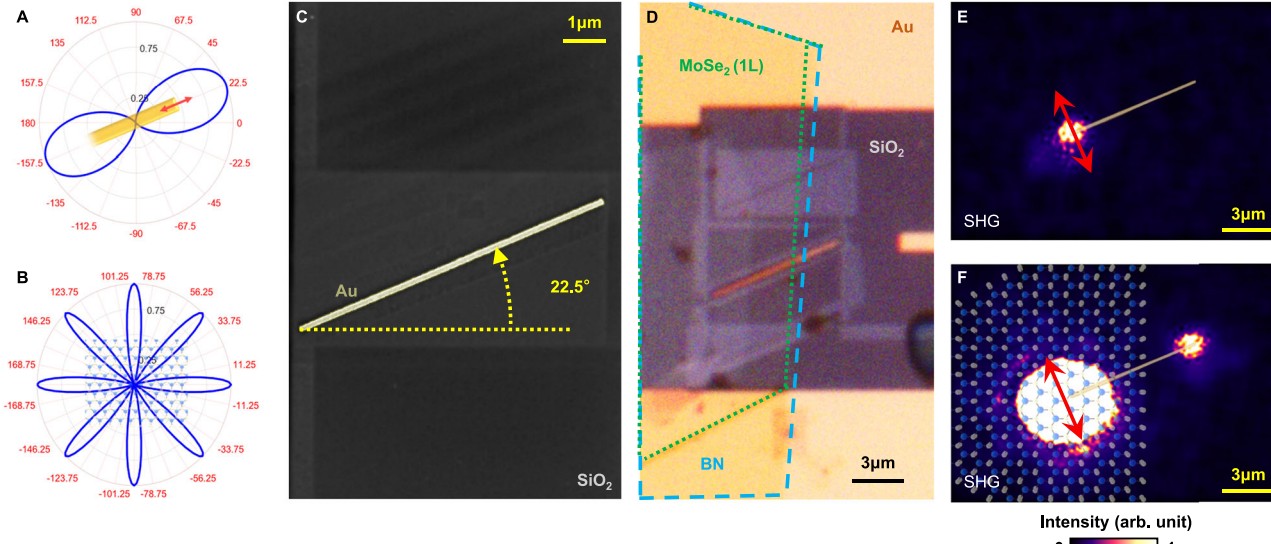

**Fig. 3 | Fabrication and performance of the MoSe₂-plasmonic hybrid single wire. A** Polar plot of output intensity as a function of polarization angle for the reference plasmonic single nanowire circuit; the nanowire orientation is rotated to 22.5°. **B** The polar plot of parallel component SHG intensity against the polarization angle of the fundamental frequency input laser. The MoSe₂ lattice is artistically presented: blue/yellow spheres represent molybdenum/selenium atoms, respectively. **C** The SEM image of the reference plasmonic single nanowire. **D** The optical microscope image of the MoSe₂-plasmonic single-wire circuit. The blue dashed lines indicate the region covered by hBN, while the region outlined by green dashed lines presents MoSe2. The SHG images for the reference (**E**) and MoSe₂-covered (**F**) single-wire circuits. The MoSe₂ lattice is artistically presented in (**F**). The red bidirectional arrows indicate the polarization angle of the input laser.

the polarization-selective router is provided in Fig. 2A. The input port of the plasmonic router is covered by a MoSe₂ monolayer with a 10 nm hexagonal boron nitride (hBN) encapsulation layer on top, as shown in Fig. 2B. The hBN/MoSe2 (from top to bottom) heterostructures are stacked in a vertical arrangement and then transferred onto TWTLs or plasmonic single-wire transmission lines. Details regarding the sample preparation are provided in Supplementary Note 1. The experiments are performed using a home-built two-color dual-confocal microscope. The fundamental frequency laser is an Er-doped mode-locked fiber laser centered at 1560 nm (Menlo Systems T-Light, producing 56 fs pulses at 80 MHz repetition rate). The linearly polarized laser beam is focused by a NIR long working distance 100x objective lens with $NA = 0.85$ (Olympus LCPLN100XIR) to excite the MoSe₂. SHG signals are collected by a visible 100x objective lens with $NA = 0.9$ (Olympus MPLFLN100X) and imaged by an electron-multiplied charge-coupled device (EMCCD, Andor iXon 897U-CS0-EXF). The detailed experimental setup is provided in Supplementary Fig. 3.

Figure 2C, D represents our experimental results on the coherent control of the MoSe₂-SHG in our nonlinear router: with a fundamental laser polarization angle tuned to $\varphi_\omega = 11.25°$ (Fig. 2C), all SHG SPPs are routed to the upper output port. Conversely, in Fig. 2D, SHG SPPs are routed to the bottom output port when the fundamental laser polarization angle is switched to $\varphi_\omega = -11.25°$. These results qualitatively confirm the TMD-induced SHG can be selectively routed by simple controls to the excitation laser polarization angle. The laser power employed is adjusted to ensure that the achieved polarization-routed SPPs are solely derived from the SHG signal generated by MoSe₂: an identical plasmonic router without MoSe₂ serves as the reference. No SHG signals can be detected in the reference plasmonic router with the same laser power, as shown in Fig. 2E, F. The details regarding other polarization angles' performance are provided in Supplementary Fig. 5.

For quantitative analysis, the routing extinction ratio is investigated using 2-D line intensity plots: the blue trace in Fig. 2G depicts the SHG intensity summed over 20 × 20 image pixel areas centered along the blue dashed line in Fig. 2C, while the orange trace is derived from the dashed line in Fig. 2D. The widths of blue and orange rectangles

correspond to the region (130 nm width) of the two output ports individually. Therefore, the experimental routing extinction ratios can be calculated as follows: For the 11.25 degrees case (routing to the top port), an extinction ratio of 16.50 dB is obtained by taking the area ratio of the two blue pattern-filled rectangles (centered at −1.5 μm divided by the one centered at 1.5 μm). For routing to the bottom port case, an extinction ratio of 14.86 dB is obtained by taking the area ratio of the two orange filled rectangles (centered at 1.5 μm divided by the one centered at −1.5 μm). On the other hand, Fig. 2H shows the log–log plots of the measured SHG output intensities as a function of input laser power for MoSe₂-plasmonic hybrid router. A slope of 2 directly reveals that the main signal is of second-order. Since only the SHG from TMD possesses the unique polarization relation[30], it is confirmed the signals being routed in this work are attributed to SHG rather than two-photon photoluminescence. Detailed descriptions are provided in Supplementary Fig. 6. Such high routing extinctions provide direct evidence to the signal coherence being preserved through various processes: (1) the fundamental laser field being converted to SHG in the MoSe₂ monolayer, (2) SHG received through the plasmonic antenna, (3) simultaneous excitation of both plasmonic TWTL modes, and lastly, (4) interference between the two modes to finally allow the desired selective routing.

## Characterization of the SHG enhancement

In order to gain further insights into the polarization dependence and SHG enhancement of such TMD-plasmonic hybrid circuit, we simplified the structural design to a plasmonic single-wire transmission line. A single plasmonic nanowire supports a single SPP mode, excited only when the optical field is polarized parallel to the wire orientation. The relationship between the polarization angle of the incident laser and the intensity of the nanowire output is displayed in Fig. 3A. Here the single nanowire is aligned to 22.5°. Figure 3B shows the relationship between the SHG intensity ($I_{\text{TMD},2\omega,x}$ component in Supplementary Fig. 1E) and the polarization angle of the fundamental frequency laser for a MoSe₂ monolayer. Details regarding the mathematical expressions of the relative rotation angle from MoSe₂ to the single nanowire are provided in Supplementary Fig. 2. We compare the SHG output

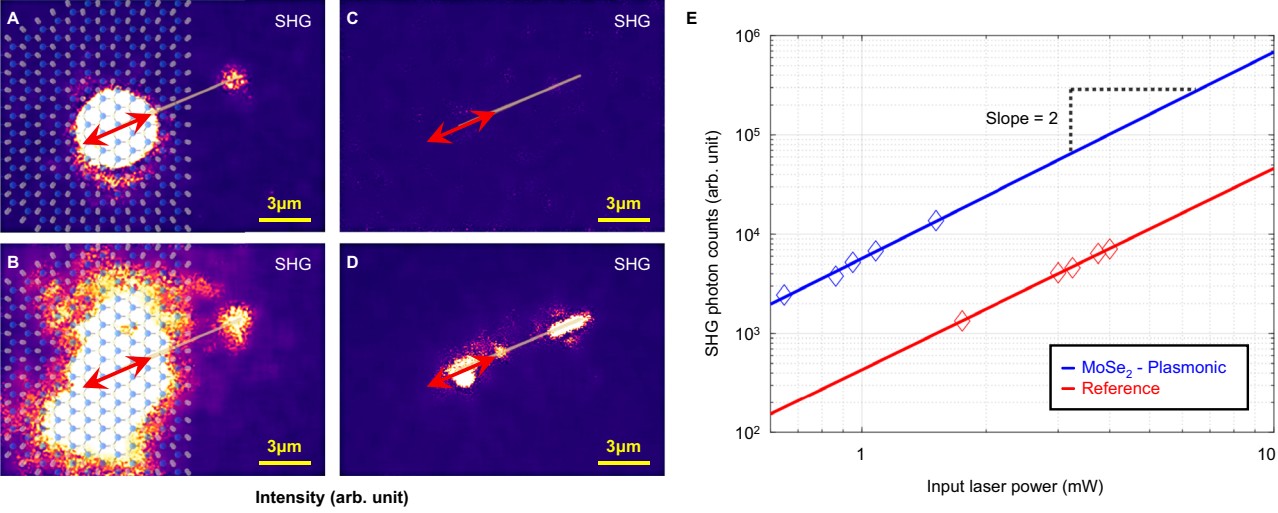

**Fig. 4 | Performance comparison in MoSe2-plasmonic circuit and pure plasmonic circuit.** SHG images of the MoSe$_2$-plasmonic (**A**, **B**) and the reference (**C**, **D**) circuits. The laser is polarized parallel to the nanowire direction, represented by red bidirectional arrows. The MoSe$_2$ lattice is artistically presented in (**A**) and (**B**). The laser powers for (**A** and **B**) are 0.75 mW and 1.5 mW, respectively. The laser powers for (**C**, **D**) are 0.75 mW and 3 mW, respectively. **E** The log–log plot of the SHG photon counts for both circuits as a function of input laser power. The experimental data are presented as symbols; fittings are depicted as lines. Direct comparison reveals the MoSe$_2$-plasmonic hybrid circuit gives rise to higher SHG conversion efficiency.

characteristics between two single-wire circuit samples: one without (reference) and one with the integrated MoSe$_2$ monolayer.

Figure 3C shows the SEM image of the fabricated reference plasmonic single-wire circuit. On the other hand, Fig. 3D shows the optical microscope image of MoSe$_2$ covered nanowire circuit. An hBN encapsulating layer is added to protect the MoSe$_2$ monolayer. We purposely excited both circuits with the laser polarized perpendicular to the nanowire ($\varphi_\omega = 112.5°$) for direct comparisons. The SHG intensities of the reference and the MoSe$_2$-covered single nanowire circuits are displayed as Fig. 3E, F, respectively. The MoSe$_2$-covered hybrid circuit in Fig. 3F shows a strong SH signal at the output port. On the other hand, no output SHG can be detected at the output port for the reference circuit, even in the case where Fig. 3E is captured with the laser power being doubled as compared to Fig. 3F. We note strong SHG can be observed at the input of the reference nanowire due to localized surface plasmon (LSP). However, the LSP-induced SHG is too weak to serve as SPP source for the plasmonic nanowire. These results provide decisive indications that the integration of TMD onto plasmonics yields much favorable and efficiency-enhanced functioning nonlinear photonic nanocircuits.

In the current work, the proposed material hybridization not only offers unique polarization routing but also large enhancement to the SHG conversion efficiency. The circuit design discussed in Fig. 3 is further used to quantify the nonlinear conversion efficiency enhancement. We compare the SHG power dependence obtained from a MoSe$_2$-plasmonic versus a reference pure plasmonic single-wire circuit (Fig. 4), respectively. For these experiments, both circuits are excited with the laser being polarized parallel to the nanowire ($\varphi_\omega = 22.5°$). This laser polarization angle allows the excitation of fundamental SPP, which in turn could be converted into second-harmonic SPP during the propagation in the reference nanowire circuit[31]. Comparisons between Fig. A, C reveal that with an incident laser power of 0.75 mW, the MoSe$_2$-plasmonic hybrid circuit produces a clear SHG signal at the output port, while the reference single wire exhibits a negligible output signal. Figure 4B shows the result of SHG image when the input laser power is increased to 1.5 mW for the MoSe$_2$-plasmonic circuit, where the SHG signals are enhanced by a factor of four as compared to Fig. 4A, confirming the power relation for a second-order nonlinear effect. To avoid damaging the MoSe$_2$

monolayer, we limited the maximum laser excitation power to 1.5 mW for the hybrid circuit.

In contrast, the reference plasmonic circuit requires an input laser power greater than 1.5 mW to provide detectable SHG signals. Figure 4D shows the reference circuit under a laser power of 3 mW. Figure 4E shows the log–log plots of the measured SHG output intensities as a function of input laser power for both circuits. For the reference circuit, the input laser directly excites the device input port. On the other hand, the partial reflection due to the existence of the hBN layer for the MoSe$_2$-plasmonic circuit was taken into consideration. We evaluated a 13.6% reflection to the laser power based on assessing the hBN refractive index at 1560 nm as 2.17[32]. Direct comparison between the two traces reveals the MoSe$_2$-plasmonic hybrid circuit exhibits a 13.8-fold enhancement of the SHG signals as compared to the reference plasmonic circuit. These findings accentuate the potential of MoSe$_2$ in enhancing nonlinear optical responses in plasmonic circuits and devices.

In summary, this work demonstrates a direct integration of a MoSe$_2$ monolayer to a polarization-selective plasmonic router. Such a hybrid TMD-plasmonic circuit combines the niches of two different material platforms in several aspects: (1) the TMD provides greatly enhanced nonlinear optical conversion efficiency; (2) the resulting SHG from the MoSe$_2$ monolayer remains to be coherent even after being coupled into the plasmonic nano-router; and (3) the TMD-generated SHG polarization dependence is seamlessly interweaved with the unique polarization-selective routing capability of the plasmonic router to enable coherent control. A simple control over the linear polarization angles of the excitation laser enables easy access to the second-harmonic SPP routing direction, achieving a routing extinction ratio as high as 16.50 dB. We experimentally reveal such material integration also leads to an extraordinary 13.8-fold enhancement in SHG conversion efficiency. In our current proof-of-concept experiment, the maximum coupling efficiency into the router is limited by the area mismatch between the antenna and the focused laser beam. Despite the mismatch in the incident coupler area of the router, the coupling efficiency could be further enhanced in the future with a new antenna design. Our current work paves the foundation to a fully integrated nonlinear photonic circuit, where the excitation laser sources, such as, quantum dot sources[33], as well as photodetectors made from two-dimensional materials, are completely embedded within such nanocircuitry.

## Methods

### Plasmonic circuit fabrication method

Scanning electron microscope images were taken with a beam voltage of 15 kV and current of 25 pA. Two settings were used for focused ion beam milling (FEI Helios): For fine structures such as the digital integrated circuits, an acceleration voltage of 5 kV and a beam current of 7.7 pA were used. For large-area milling, 5 kV and 80 pA were used to gain the etching speed.

### MoSe$_2$ fabrication method

In this study, the two-dimensional materials were prepared using the mechanical exfoliation method. The thicknesses of hBN flakes were determined using atomic force microscopy, while the presence of MoSe$_2$ monolayers was confirmed through optical microscopy and verified by their photoluminescence spectra.

The creation of the hybrid plasmonic nanocircuitry involved employing a dry transfer technique to vertically assemble different materials. Specifically, we stacked hBN/MoSe$_2$ heterostructures in a vertical arrangement and then transferred them onto TWTLs or plasmonic single-wire transmission lines. To achieve polarization-selective routing of SHG, careful management of the crystallographic alignment of MoSe$_2$ with respect to the plasmonic structure was implemented during the transfer processes.

## Data availability

All data generated during the current study are available from the corresponding authors upon request.

## Code availability

All code used in this study is available from the corresponding authors upon request.

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

## Acknowledgements

This work was supported by the National Science and Technology Council under Grant 109-2112-M-007-031-MY3 (CBH) and Grant 112-2112-M-007-021-MY3 (CBH).

## Author contributions

P.Y.W. performed numerical simulation, plasmonic device fabrication, optical measurement, and prepared the figures. W.Q.L. fabricated the TMD. C.H.L. and C.B.H. initiated the concept, supervised the project, and wrote the manuscript.

## Competing interests

The authors declare no competing interests.
