## [Peer Review File · Nature Communications]

Coherent control of enhanced second-harmonic generation in a plasmonic nanocircuit using a transition metal dichalcogenide monolayerEditorial Note: Parts of this Peer Review File have been redacted as indicated to remove third-party material where no permission to publish could be obtained.

REVIEWER COMMENTS

Reviewer #1 (Remarks to the Author):

In this work, the authors have performed the direct integration of transitional metal dichalcogenide (TMD) onto a plasmonic nanocircuitry, where the authors have demonstrated the selective routing of second-harmonic generation (SHG) signals from a MoSe₂ monolayer within the plasmonic circuit via controlling the polarization angle of the input laser. The experiments have demonstrated a routing extinction ratio of more than 14.86 dB. The presented work is of strong interest to the research community. Would like to suggest a minor revision of this manuscript, before its acceptance for publication in Nature Communications.

1. Figure 2C and Figure 2D present the key research work, where the polarization angle of 11.25 degree is able to control the SHG emission. Besides 11.25 degree, is there any other polarization angle capable of achieving similar performance in controlling the routed SHG emission.
2. In the abstract, when mentioning on the SHG emission enhancement, would like to suggest the authors to change “1380.27%” to “13.8-fold”.
3. Image quality. For some figures (e.g., Figure 2A-2D), if we zoom in, we can see the pixelated images. Would like to suggest the authors to improve these figures, especially Figure 2B. In addition, in Figure 2B, why there is a dark rectangular shape region around the plasmon router? In Figure 2A, it is good to label the substrate material and the plasmonic waveguide material.
4. In Figure 4E on the log-log plot of the SHG photon, the number of measurement points for the reference is only 2. Would like to suggest the authors to add at least another 2-3 points for the reference and 1-2 more point for “MoSe₂-plasmonic”.

Reviewer #2 (Remarks to the Author):

The paper reports on SHG routing in a hybrid 2D semiconductor plasmonic structure. An hBN encapsulated or capped MoSe₂ monolayer is transferred onto pre-fabricated plasmonic waveguide. The TMD plasmonic structure is excited in a gap region of a two wire transmission line (TWTL), and then they measure the SHG intensity that is funneled to two different output ports by SPPs. The waveguide is designed so that it has polarization sensitive routing, and they show that by varying the excitation polarization, they can preferentially route the generated SHG out of a specific output port, which I believe is a novel effect for TMD generated SHG. The primary result is a demonstration of this routing effect, but the effect itself is not surprising, and I do

not feel that the result or paper quality is worthy of publication in Nature Communications. They show the routing result on one device and report the routing efficiency on what appears to be two measurements, shown in Figure 2. The authors should have been able to measure the routing efficiency as they vary the excitation polarization angle, which I find to be a strange omission.

Other comments:

I assume in the Figure 2 routing experiment, they performed power dependence to confirm that the signal is SHG. That should be shown.

Is the TMD really hBN encapsulated or just capped on top. This wasn't clear.

Was the routing demonstrated on a second device?

How do the authors distinguish SHG from two-photon PL?

The paper is not organized well. I believe Figure 1 mostly shows theory, but the text should clearly state this.

What are the physical dimensions of the waveguide? Is the gap size crucial?

What do is the loss as the SPP propagates through the curved channel?

All figure captions need to be improved for clarity.

Exactly what filter was used to separate the SHG from the excitation laser?

One of the figures of merit reported 1380% appears to be the ratio of the SHG generated by the TMD-plasmonic sample to the SHG generated by a bare waveguide. This seems like a strange figure of merit. The authors should compare the SHG efficiency of a bare TMD to that of the TMD-plasmonic structure.

Reviewer #3 (Remarks to the Author):

The authors studied second-harmonic generation in a TMD monolayer which is integrated with a plasmonic nanocircuit to allow the selective routing of the SHG signal by simply tuning the polarization of the input laser. Compared to a reference circuit without TMD, the integration of TMD leads to a SHG enhancement of 1380.27%, while the routing extinction ratio is estimated to be 14.86 dB and 16.50 dB. Although based on the data shown in Fig. 2, the in-coupling efficiency (from SHG generated in TMD to SHG coupled to plasmonic waveguide) seems to be not high, the out-coupled SHG signal at the end of the plasmonic router is still clear. As also mentioned by the author, so far the SHG studies in TMD materials are usually detected directly in the monolayer region, while this paper shows the possibility of

controllable routing of SHG signal for applications in nonlinear optical circuit. In terms of both novelty and importance, I rate the submitted paper highly. All the experiments are carefully designed which can well support their conclusion. There are only some minor issues such as the data analysis can be improved a bit to make it clearer to the reader. Thus, I would like to suggest the publication of this manuscript in Nature Communications after some minor revisions.

Comment 1:

The claim in the last sentence of the third paragraph "However, it's worth noting that TMD-generated SHGs have, so far, only been reported in a mechanism involving free-space excitation and free-space collection" should be a bit more careful. For example, the Ref. 4 in the reference list also reported remote excitation of SHG in monolayer TMD via SPP generated in Ag nanowire. They did not see out-coupling of SHG signal at the nanowire end, may be due to the short wavelength of their SHG signal, which limits its propagation.

Comment 2:

In Fig. 2, the routing extinction ratios are reported as 16.50 dB and 14.86 dB, respectively. How the two numbers were obtained? What is the ratio if we divide the SHG intensity scattered from the waveguide end by the total SHG intensity excited in the TMD monolayer? In this way, the ratio may be further lowered. Can the author briefly discuss the in-coupling efficiency and propagation loss, separately?

Comment 3:

Related to the above comment, in Fig. 4, when we compare panel a and d, although the SHG signal generated in TMD monolayer region looks higher than SHG signal generated in the plasmonic nanowire, at the waveguide end, the SHG signal scattered out in panel a is weaker than that of panel d. This again suggests that a lot of SHG signal generated in the TMD monolayer is wasted which does not efficiently couple into the waveguide. For this point, the author may pay more attention in the future studies. Some antennas may be designed and integrated with the waveguide.

Comment 4:

The author used a 1560 nm laser, and the generated SHG of 780 nm matches well with the exciton resonance of MoSe₂. Is it possible that there is also contribution from two-photon excited PL in these samples? Have the authors checked the SHG spectrum or tuned the laser wavelength to check the reproducibility?

Replies and Actions Taken to Reviewer Comments

We highly appreciate the three Reviewers' comments and efforts in making our manuscript stronger. Below please find the reviewers' comments, followed by our **replies** in **green texts**, and our **actions taken** in order to address their comments in **blue**. To track our changes, please kindly refer to the highlighted version of our revised manuscript and Supplementary Information, combined as one single file to facilitate your review.

Replies to Reviewer 1

Comments:

In this work, the authors have performed the direct integration of transitional metal dichalcogenide (TMD) onto a plasmonic nanocircuitry, where the authors have demonstrated the selective routing of second-harmonic generation (SHG) signals from a MoSe₂ monolayer within the plasmonic circuit via controlling the polarization angle of the input laser. The experiments have demonstrated a routing extinction ratio of more than 14.86 dB. The presented work is of strong interest to the research community. Would like to suggest a minor revision of this manuscript, before its acceptance for publication in Nature Communications.

Reply: We thank this reviewer for the very positive recognitions. Below please find our point-by-point replies to your questions.

1. Figure 2C and Figure 2D present the key research work, where the polarization angle of 11.25 degree is able to control the SHG emission. Besides 11.25 degree, is there any other polarization angle capable of achieving similar performance in controlling the routed SHG emission.

Reply: Thanks for pointing out this issue. According to the polarization dependence described in Figure 1, routing to the top port is expected at 11.25 degrees as well as $90 + 11.25 = 101.25$ degrees. On the other hand, routing to the bottom port is expected at -11.25 degrees and $90 - 11.25 = 78.75$ degrees. We have experimentally confirmed these and the results are provided for you below:

Action taken: Please refer to the revised **supplementary information**, where we have added a section entitled “Other routing polarization angles” to address your comment.

2. In the abstract, when mentioning on the SHG emission enhancement, would like to suggest the authors to change “1380.27%” to “13.8-fold”.

Reply: Thank you for the suggestion. We have modified the SHG enhancement to “13.8-fold” per your suggestion.

Action taken: The enhancement values on Pages 1, 3, 11, and 12 have been modified to “13.8-fold”. Please kindly refer to the highlighted version of the revised manuscript.

3. Image quality. For some figures (e.g., Figure 2A-2D), if we zoom in, we can see the pixelated images. Would like to suggest the authors to improve these figures, especially Figure 2B. In addition, in Figure 2B, why there is a dark rectangular shape region around the plasmon router? In Figure 2A, it is good to label the substrate material and the plasmonic waveguide material.

Reply: Thank you for your comment. We have tried to improve the image qualities. The dark rectangular regions in Figures 2A and 2B are the SiO₂ substrate, as a result of focused-ion beam milling of gold in defining our plasmonic router. Thank you for the suggestion, we have labeled the corresponding materials in revised Figures 2A and 2B as well as Figure 3D.

Action taken: The labels indicating different materials have been added to Figure 2B on page 8 and Figure 3D on page 10 in the revised manuscript. Caption to Figure 2A has also been revised to address the corresponding materials.

4. In Figure 4E on the log-log plot of the SHG photon, the number of measurement points for the reference is only 2. Would like to suggest the authors to add at least another 2-3 points for the reference and 1-2 more point for “MoSe₂-plamonic”.

Reply: We re-performed the measurements per your nice suggestion. For the MoSe₂-plasmonic structure, we used five incident powers of 0.75mW, 1mW, 1.1mW, 1.25mW, and 1.75mW. We limited the laser power below 2 mW in order to avoid damaging the TMD. For the reference structure, we sampled at five incident powers of 1.75mW, 3mW, 3.25mW, 3.5mW and 3.75mW. The updated log-log plot is reproduced here for your convenience.

Action taken: We updated Figure 4E on page 11 with the new experimental data. Each trace is represented with 5 experimental measured points.

Replies to Reviewer 2

Comments:

The paper reports on SHG routing in a hybrid 2D semiconductor plasmonic structure. An hBN encapsulated or capped MoSe₂ monolayer is transferred onto pre-fabricated plasmonic waveguide. The TMD plasmonic structure is excited in a gap region of a two wire transmission line (TWTL), and then they measure the SHG intensity that is funneled to two different output ports by SPPs.

The waveguide is designed so that it has polarization sensitive routing, and they show that by varying the excitation polarization, they can preferentially route the generated SHG out of a specific output port, which I believe is a novel effect for TMD generated SHG.

The primary result is a demonstration of this routing effect, but the effect itself is not surprising, and I do not feel that the result or paper quality is worthy of publication in Nature Communications. They show the routing result on one device and report the routing efficiency on what appears to be two measurements, shown in Figure 2. The authors should have been able to measure the routing efficiency as they vary the excitation polarization angle, which I find to be a strange omission.

Reply: We appreciate your questions and comments. We have addressed all of your points and we are indebted to your efforts in helping us to further strengthen our manuscript.

1. I assume in the Figure 2 routing experiment, they performed power dependence to confirm that the signal is SHG. That should be shown.

Reply: Thank you for this suggestion. The power relation shows an evident slope of 2.

Action taken: We have added the power relation as Figure 2H and have revised the corresponding figure caption to Figure 2. Please kindly refer to the highlighted revised manuscript.

2. Is the TMD really hBN encapsulated or just capped on top. This wasn't clear.

Reply: Thanks for the question. The process of our sample preparation was already described in the Supplementary Information under “MoSe₂ fabrication method” section. However, we will make it more clear per your suggestion.

Action taken: We have explicitly stated the encapsulation issue in our revised manuscript, starting on the bottom of page 6 and are reproduced here for your convenience: “The hBN/MoSe₂ (from top to bottom) heterostructures are stacked in a vertical arrangement and then transferred onto TWTLs or plasmonic single-wire transmission lines. Details regarding sample preparation is provided in the **supplementary information.**”

3. Was the routing demonstrated on a second device?

Reply: Absolutely. During our investigations, we have demonstrated selective routing using multiple TMD materials on several plasmonic circulators/routers. Furthermore, our experiments have shown extremely high reproducibility in our response to question 4 raised by Reviewer 1. Below please find our experimental routing results for three different TMD materials:

Figures R1A and **R1B** demonstrate the selective routing of SHG using CVD-prepared WSe₂ at ± 11.25 degrees in a plasmonic circulator (device explained in Ref. 25). **Figures R1C** and **R1D** demonstrate the routing of SHG from exfoliated WSe₂ in a plasmonic circulator. However, due to intrinsic material property, the resulting SHG from WSe₂ are weaker as compared to MoSe₂ (presented in the manuscript, shown here as **Figures R1E** and **R1F**). In this paper, we therefore present the results using MoSe₂. It is clear that the polarization routings are universally demonstrated across different TMD materials on different plasmonic circulators/routers.

Figure R1: (A), (B) the routing of SHG from CVD-prepared WSe₂. (C), (D) the routing of SHG from exfoliated WSe₂. (E), (F) the routing of SHG from exfoliated MoSe₂.

4. How do the authors distinguish SHG from two-photon PL?

Reply: This is a great question. We address your comment from three perspectives:

- (1) Although SHG and two-photon PL from TMD have the possibility to be simultaneously excited, it is known in literature that the intensity of two-photon PL is much weaker as compared to SHG^{1,2}.
- (2) SHG signal from TMDs have a unique polarization relationship³ to the fundamental frequency input laser, as shown in **Figure R2A** (adapted from Nature Nanotech. **10**, 407 (2015)) below. Also shown in **Figure R2A**, two-photon PL, being incoherent in nature, does not possess such polarization relationship³. Therefore, we performed the TMD polarization dependence studies using spectral measurements (by Andor Shamrock 193i spectrometer with Andor Newton CCD) to ensure that the signal from MoSe₂ is SHG-dominated rather than two-photon PL, as shown in **Figure R2B**. The polarization insensitive trace (two-photon PL, orange trace), is acquired by tuning the polarization analyzer so that the SHG signal is minimized, and is found to be 15.6 dB weaker as compared to the SHG signal (blue trace). This finding is in great accord to our achieved experimental routing extinction ratio.
- (3) Related to your question 9, in our experimental setup, a band-pass filter (Thorlabs FL780-10) is used to ensure the collected signals are centered at the SHG peak at 780 nm.

Figure R2: (A) Resonant SHG and two-photon PL intensity (under non-resonant excitation at 0.88 eV) parallel to the incident laser polarization as a function of crystal angle, where the black line shows the expected SHG polarization dependence. Adapted from [Nature Nanotech. **10**, 407 (2015).] (B) Polarization dependence of SHG intensity from MoSe₂ monolayer through spectral measurements. The blue trace represents the spectrum of when SHG is maximum, while the orange trace denotes the two-photon PL spectrum (when the polarization analyzer minimized the SHG transmission).

Action taken: We have added a new section entitled “Distinguishing SHG from two-photon PL” to the revised **supplementary information** per your comment. Also, we added a brief discussion on page 8 to address this issue, reproduced here for your convenience: “Since only the SHG from TMD possess the unique polarization relation, it is confirmed the signals being routed in this work are attributed to SHG rather than two-photon photoluminescence. Detailed descriptions are provided in **supplementary Figure S6.**”

5. The paper is not organized well. I believe Figure 1 mostly shows theory, but the text should clearly state this.

Reply: We respect your opinion. However, after full consideration, we still believe it would be more beneficial for the readers to grasp the underlying working principles first, rather than receiving a full load of the experimental results upfront. In addition, please note in the section where Figure 1 is presented, we explicitly stated multiple times that Figure 1 serves as an illustration to the design concept: (1) The section title "Circuit working principle" clearly indicates that the upcoming contents are conceptual explanations of the working principle. (2) The first sentence of the paragraph states, "Figure 1 schematically illustrates the working principle of our hybrid nonlinear router: ..." providing a comprehensive explanation that Figure 1 is intended as a theoretical representation.

Action taken: We added the wording “Working principle.” to the revised Figure 1 caption, to clearly indicate Figure 1 is meant for conceptual explanations per your request.

6. What are the physical dimensions of the waveguide? Is the gap size crucial?

Reply: Thank you for raising this question. We address this issue with the details as follows:

The physical dimensions of the waveguide

As shown in the figure (a) below, section S1 is a TWTL structure that couples laser input into surface plasmon polaritons (SPP). The input source polarization provides SPP modal selectivity. The separation between the two nanowires in section S2 is gradually increased so the SPP fields on the two nanowires are no longer coupled and thus enabling routing. Therefore, the gap size is a crucial design parameter. Section S3 is used as a buffer region for bending loss control.

(b) The cross-sectional view of the device. Basic TWTL parameters include wire width W_{wire} , gap width W_{gap} , and height H . In our designs we used $H = 60 \text{ nm}$, $W_{\text{wire}} = 130 \text{ nm}$, and $W_{\text{gap}} = 80 \text{ nm}$ in sections S1.

(c) For section S2, the nanowires are designed according to the quadratic Bézier curve through $P(t) = (1 - t)^3 P_0 + 3t(1 - t)^2 P_1 + 3t^2(1 - t) P_2 + t^3 P_3$. Such curve is defined through the designations of the four spatial points ($P_0 \sim P_3$). The spatial coordinates of the four points are simplified through the defining the sbend_W and sbend_L of the bended nanowire.

(d) Our router circuit has values: Input_L = 400 nm, Output_L = 400 nm, sbend_W = 3000 nm, sbend_L = 5200 nm respectively. An additional coefficient a within ranges of 0~1 provide design flexibility and optimization (here, a is 0.5).

Action taken: We have added a new section entitled “Design parameters for the plasmonic TWTL router circuit” to the revised **supplementary information** per your comment. We also added a sentence “Details of TWTL design are provided in the **supplementary information.**” to the end of paragraph 1 on page 5. Please kindly refer to the highlighted revised manuscript.

7. What is the loss as the SPP propagates through the curved channel?

Reply: Reviewer 3 also asked a similar question that not only involved bending loss, but also about the calculation of the in-coupling efficiency and propagation loss. Therefore, please kindly refer to our responses to Reviewer 3, question 2 for details.

8. All figure captions need to be improved for clarity.

Reply: Thank you for the suggestion. We have improved all figure captions.

Action taken: We have added explanations regarding the labels in **Figures 1-4** (Pages 6, 8, 10 and 12) in the revised manuscript.

9. Exactly what filter was used to separate the SHG from the excitation laser?

Reply: We first use a short pass filter (Thorlabs FESH0900) to eliminate the laser excitation, followed by a band-pass filter (Thorlabs FL780-10) to ensure that the recorded images are centered at 780 nm. The corresponding spectra are provided for your reference in **Figures R3A** and **R3B**, respectively.

Figure R3: (A) The transmission spectrum of the short pass filter (Thorlabs FESH0900). (B) The transmission spectrum of the band-pass filter (Thorlabs FL780-10). Data adapted from Thorlabs website.

Action taken: The filter model information is provided in the captions to **Supplementary Figure S3**.

10. One of the figures of merit reported 1380% appears to be the ratio of the SHG generated by the TMD-plasmonic sample to the SHG generated by a bare waveguide. This seems like a strange figure of merit. The authors should compare the SHG efficiency of a bare TMD to that of the TMD-plasmonic structure.

Reply: We respect your opinion. However, the emphasis is achieving SHG signal enhancement for a photonic nanocircuitry. Therefore, we believe that our design in **Figure 3** is reasonable, and the SHG enhancement should be referenced to a pure plasmonic structure.

Replies to Reviewer 3

Comments:

The authors studied second-harmonic generation in a TMD monolayer which is integrated with a plasmonic nanocircuit to allow the selective routing of the SHG signal by simply tuning the polarization of the input laser. Compared to a reference circuit without TMD, the integration of TMD leads to a SHG enhancement of 1380.27%, while the routing extinction ratio is estimated to be 14.86 dB and 16.50 dB. Although based on the data shown in Fig. 2, the in-coupling efficiency (from SHG generated in TMD to SHG coupled to plasmonic waveguide) seems to be not high, the out-coupled SHG signal at the end of the plasmonic router is still clear. As also mentioned by the author, so far the SHG studies in TMD materials are usually detected directly in the monolayer region, while this paper shows the possibility of controllable routing of SHG signal for applications in nonlinear optical circuit. In terms of both novelty and importance, I rate the submitted paper highly. All the experiments are carefully designed which can well support their conclusion. There are only some minor issues such as the data analysis can be improved a bit to make it clearer to the reader. Thus, I would like to suggest the publication of this manuscript in Nature Communications after some minor revisions.

Reply: Thank you very much for your recognition. We have addressed each of your questions below.

1. The claim in the last sentence of the third paragraph “**However, it's worth noting that TMD-generated SHGs have, so far, only been reported in a mechanism involving free-space excitation and free-space collection**” should be a bit more careful. For example, the Ref. 4 in the reference list also reported remote excitation of SHG in monolayer TMD via SPP generated in Ag nanowire. They did not see out-coupling of SHG signal at the nanowire end, may be due to the short wavelength of their SHG signal, which limits its propagation.

Reply: Thanks for the suggestion. We have revised the sentence.

Action taken: The claim in the last sentence of the third paragraph on page 3 is modified to “However, it's worth noting that TMD-generated SHGs have, so far, predominantly been reported in a mechanism involving free-space excitation and free-space collection”

2. In Fig. 2, the routing extinction ratios are reported as 16.50 dB and 14.86 dB, respectively. How the two numbers were obtained? What is the ratio if we divide the SHG intensity

scattered from the waveguide end by the total SHG intensity excited in the TMD monolayer? In this way, the ratio may be further lowered. Can the author briefly discuss the in-coupling efficiency and propagation loss, separately?

Reply: In general, for optical communication components such as routers and circulators, the extinction ratio is defined as the intensity ratio between the intended and unintended ports. This value is typically expressed in decibels (dB). A high extinction ratio ensures superior signal quality, reduced errors and interference, ensuring reliable data transmission.

Calculating the routing extinction ratios

In Figure 2(G), the two output port locations are marked by the rectangles (400 nm width that equals to the nanowire width). For the 11.25 degrees case (routing to the top port), the extinction ratio is obtained by taking the area ratio of the two blue slant-filled rectangles (centered at $-2.5 \mu\text{m}$ divided by the one centered at $2.5 \mu\text{m}$). For routing to the bottom port case, the extinction ratio is obtained by taking the area ratio of the two orange filled rectangles (centered at $2.5 \mu\text{m}$ divided by the one centered at $-2.5 \mu\text{m}$).

Action taken: The explanation on page 8 regarding the calculation procedure for routing extinction is revised, and is reproduced here for your convenience: “The widths of blue and orange rectangles correspond to the region (400 nm width) of two output ports individually. Therefore, the experimental routing extinction ratios can be calculated as follows: For the 11.25 degrees case (routing to the top port), an extinction ratio of 16.50 dB is obtained by taking the area ratio of the two blue pattern-filled rectangles (centered at $-2.5 \mu\text{m}$ divided by the one centered at $2.5 \mu\text{m}$). For routing to the bottom port case, an extinction ratio of 14.86 dB is

obtained by taking the area ratio of the two orange filled rectangles (centered at $2.5 \mu\text{m}$ divided by the one centered at $-2.5 \mu\text{m}$).”

The in-coupling efficiency and propagation loss

In our current proof of concept experiment, the free-spaced laser beam is focused by an 100x objective lens. Due to diffraction-limit and the size of the clear aperture of the objective lens, the focused beam size is roughly $1 \mu\text{m}^2$. This creates a seemingly mismatch to the incident coupling area of the router being around $0.26 \mu\text{m}^2$ and therefore limits the maximum coupling efficiency.

In a router design, we considered propagation loss as well as bending loss. To verify these considerations, we conducted analyses for each case by simulations. The curved design of the router is based on quadratic Bézier curve (for details please refer to our replies to Reviewer 2’s question 6). Using such equation, the length of curve was calculated and numerically evaluated using Lumerical FDTD. Two cases were compared: one is a curved single wire (Figure A below) which resembles our router design and the other is a single straight wire (Figure B below). Both structures have identical lengths. The intensity variation along the y-axis for the output is illustrated in Figure C, revealing an intensity loss of 29.93% attributed to the curved design.

On the other hand, we estimate a propagation loss of approximately 27.92% using the single wire with the same length as curved wire.

3. Related to the above comment, in Fig. 4, when we compare panel a and d, although the SHG signal generated in TMD monolayer region looks higher than SHG signal generated in the plasmonic nanowire, at the waveguide end, the SHG signal scattered out in panel a is weaker than that of panel d. This again suggests that a lot of SHG signal generated in the TMD monolayer is wasted which does not efficiently couple into the waveguide. For this point,

the author may pay more attention in the future studies. Some antennas may be designed and integrated with the waveguide.

Reply: We greatly appreciate your valuable input. Due to the issues discussed in the second question, the coupling efficiency is constrained by the mismatch in the antenna area to the exciting laser beam. There are two future approaches that could improve the device coupling efficiency: (1) A new input antenna structure aimed to better match the focused laser beam size. (2) Alternatively, we are in the process of integrating quantum emitters to our plasmonic devices, which might be the ultimate solution for a hybrid plasmonic circuitry. We would be more than happy to update the community when successful.

Action taken: We have updated the Conclusion section on page 13 for potential future improvements and is reproduced here for your convenience: “In our current proof of concept experiment, the maximum coupling efficiency into the router is limited by the area mismatch between the antenna and the focused laser beam. Despite the mismatch in the incident coupler area of the router. The coupling efficiency could be further enhanced in the future with a new antenna design.”

4. The authors used a 1560 nm laser, and the generated SHG of 780 nm matches well with the exciton resonance of MoSe₂. Is it possible that there is also contribution from two-photon excited PL in these samples? Have the authors checked the SHG spectrum or tuned the laser wavelength to check the reproducibility?

Reply: Thank you and this is indeed an important issue. Reviewer 2 also raised the same question in her/his comment 4. Please refer to pages 6 and 7 where we replied to Reviewer 2. Unfortunately, our excitation source is a commercially available fiber-based passively mode-locked laser (Menlo Systems T-Light), which does not allow tuning of the laser center wavelength.

Action taken: We have added a new section entitled “Distinguishing SHG from two-photon PL” to the revised **supplementary information** per your comment.

References

- 1 Granados del Águila, A. s. *et al.* Linearly polarized luminescence of atomically thin MoS₂ semiconductor nanocrystals. *ACS nano* **13**, 13006-13014 (2019).
- 2 Han, C. & Ye, J. Polarized resonant emission of monolayer WS₂ coupled with plasmonic sawtooth nanoslit array. *Nature communications* **11**, 713 (2020).

- 3 Seyler, K. L. *et al.* Electrical control of second-harmonic generation in a WSe₂ monolayer transistor. *Nature nanotechnology* **10**, 407-411 (2015).

REVIEWERS' COMMENTS

Reviewer #1 (Remarks to the Author):

Accept

Reviewer #2 (Remarks to the Author):

The manuscript has been improved significantly and is now suitable for publication.

Reviewer #3 (Remarks to the Author):

The manuscript “Coherent control of enhanced second-harmonic generations in a plasmonic nanocircuit using a transitional metal dichalcogenide monolayer” by Wu et al. demonstrates the possibility of controllable routing of SHG signal for potential applications in nonlinear optical circuit, which I consider it to be a very important result. The authors have addressed all my comments thoroughly and in a very satisfactory manner. Therefore, I would like to recommend the acceptance of this manuscript for publication in Nature Communications.